# Fair Algorithms for Multi-Agent Multi-Armed Bandits

**Safwan Hossain**
Department of Computer Science
University of Toronto
safwan.hossain@mail.utoronto.ca

**Evi Micha**
Department of Computer Science
University of Toronto
emicha@cs.toronto.edu

**Nisarg Shah**
Department of Computer Science
University of Toronto
nisarg@cs.toronto.edu

## Abstract

We propose a multi-agent variant of the classical multi-armed bandit problem, in which there are $N$ agents and $K$ arms, and pulling an arm generates a (possibly different) stochastic reward for each agent. Unlike the classical multi-armed bandit problem, the goal is not to learn the "best arm"; indeed, each agent may perceive a different arm to be the best for her personally. Instead, we seek to learn a fair distribution over the arms. Drawing on a long line of research in economics and computer science, we use the *Nash social welfare* as our notion of fairness. We design multi-agent variants of three classic multi-armed bandit algorithms and show that they achieve sublinear regret, which is now measured in terms of the lost Nash social welfare.

## 1 Introduction

In the classical (stochastic) multi-armed bandit (MAB) problem, a principal has access to $K$ arms and pulling arm $j$ generates a stochastic reward for the principal from an unknown distribution $D_j$ with an unknown mean $\mu_j^*$. If the mean rewards were known a priori, the principal could just repeatedly pull the *best arm* given by $\arg\max_j \mu_j^*$. However, the principal has no apriori knowledge of the quality of the arms. Hence, she uses a learning algorithm which operates in rounds, pulls arm $j^t$ in round $t$, observes the stochastic reward generated, and uses that information to learn the best arm over time. The performance of such an algorithm is measured in terms of its cumulative regret up to a horizon $T$, defined as $\sum_{t=1}^{T}(\max_j \mu_j^* - \mu_{j^t}^*)$. Note that this is the difference between the total mean reward that would have been achieved if the best arm was pulled repeatedly and the total mean reward of the arms pulled by the learning algorithm up to round $T$.

This problem can model situations where the principal is deliberating a policy decision and the arms correspond to the different alternatives she can implement. However, in many real-life scenarios, making a policy decision affects not one, but several agents. For example, imagine a company making a decision that affects all its employees, or a conference deciding the structure of its review process, which affects various research communities. This can be modeled by a multi-agent variant of the multi-armed bandit (MA-MAB) problem, in which there are $N$ agents and pulling arm $j$ generates a (possibly different) stochastic reward for each agent $i$ from an unknown distribution $D_{i,j}$ with an unknown mean $\mu_{i,j}^*$.

Before pondering about learning the "best arm" over time, we must ask what the best arm even means in this context. Indeed, the "best arm" for one agent may not be the best for another. A first attempt may be to associate some "aggregate quality" to each arm; for example, the quality of arm

35th Conference on Neural Information Processing Systems (NeurIPS 2021).

$j$ may be defined as the total mean reward it gives to all agents, i.e., $\sum_i \mu_{i,j}^*$. This would nicely reduce our problem to the classic multi-armed bandit problem, for which we have an armory of available solutions [1]. However, this approach suffers from the *tyranny of the majority* [2]. For example, imagine a scenario with ten agents, two arms, and deterministic rewards. Suppose four agents derive a reward of 1 from the first arm but 0 from the second, while the remaining six derive a reward of 1 from the second arm but 0 from the first. The aforementioned approach will deem the second arm as the best and a classical MAB algorithm will converge to *repeatedly* pulling the second arm, thus unfairly treating the first four agents (a minority). A solution which treats each group in a "proportionally fair" [2] manner should ideally converge to pulling the first arm $40\%$ of the time and the second $60\%$ of the time. Alternatively, we can allow the learning algorithm to "pull" a probability distribution over the arms and seek an algorithm that converges to placing probability $0.4$ on the first arm and $0.6$ on the second.

This problem of making a fair collective decision when the available alternatives — in this case, probability distributions over the arms — affect multiple agents is well-studied in computational social choice [3]. The literature offers a compelling fairness notion called the *Nash social welfare*, named after John Nash. According to this criterion, the fairest distribution maximizes the *product* of the expected utilities (rewards) to the agents. A distribution $p$ that places probability $p_j$ on each arm $j$ gives expected utility $\sum_j p_j \cdot \mu_{i,j}^*$ to agent $i$. Hence, the goal is to maximize $\mathrm{NSW}(p, \mu^*) = \prod_{i=1}^N (\sum_{j=1}^K p_j \cdot \mu_{i,j}^*)$ over $p$. One can verify that this approach on the aforementioned example indeed yields probability $0.4$ on the first arm and $0.6$ on the second, as desired. It is also interesting to point out that with a single agent ($N = 1$), the distribution maximizing the Nash social welfare puts probability 1 on the best arm, thus effectively reducing the problem to the classical multi-armed bandit problem (albeit with subtle differences which we highlight in Appendix A).

Maximizing the Nash social welfare is often seen as a middle ground between maximizing the utilitarian social welfare (sum of utilities to the agents), which is unfair to minorities (as we observed), and maximizing the egalitarian social welfare (minimum utility to any agent), which is considered too extreme [2]. The solution maximizing the Nash social welfare is also known to satisfy other qualitative fairness desiderata across a wide variety of settings [4–11]. For example, a folklore result shows that in our setting such a solution will always lie in *the core*; we refer the reader to the work of Fain et al. [4] for a formal definition of the core as well as a short derivation of this fact using the first-order optimality condition. For further discussion on this, see Sections 1.2 and 6.

When *exactly* maximizing the Nash social welfare is not possible (either due to a lack of complete information, as in our case, or due to computational difficulty), researchers have sought to achieve approximate fairness by approximately maximizing this objective [12–17]. Following this approach in our problem, we define the (cumulative) regret of an algorithm at horizon $T$ as $\sum_{t=1}^T (\max_p \mathrm{NSW}(p, \mu^*) - \mathrm{NSW}(p^t, \mu^*))$, where $p^t$ is the distribution selected in round $t$. Our goal in this paper is to design algorithms whose regret is sublinear in $T$.

## 1.1 Our Results

We consider three classic algorithms for the multi-armed bandit problem: Explore-First, Epsilon-Greedy, and UCB [1]. All three algorithms attempt to balance exploration (pulling arms only to learn their rewards) and exploitation (using the information learned so far to pull "good" arms). Explore-First performs exploration for a number of rounds optimized as a function of $T$ followed by exploitation in the remaining rounds to achieve regret $\widetilde{\mathcal{O}}\left(K^{1/3}T^{2/3}\right)$. Epsilon-Greedy flips a coin in each round to decide whether to perform exploration or exploitation and achieves the same regret bound. Its key advantage over Explore-First is that it does not need to know the horizon $T$ upfront. UCB merges exploration and exploitation to achieve a regret bound of $\widetilde{\mathcal{O}}\left(K^{1/2}T^{1/2}\right)$. Here, $\widetilde{\mathcal{O}}$ hides log factors. Traditionally, the focus is on optimizing the exponent of $T$ rather than that of $K$ as the horizon $T$ is often much larger than the number of arms $K$. It is known that the dependence of UCB's regret on $T$ is optimal: no algorithm can achieve *instance-independent* $o(T^{1/2})$ regret [18].[1]

---

[1]In instance-independent bounds, the constant inside the big-Oh notation is not allowed to depend on the (unknown) distributions in the given instance. UCB also achieves an $O(\log T)$ instance-dependent regret bound, which is also known to be asymptotically optimal [19]. For further discussion, see Section 6.

We propose natural multi-agent variants of these three algorithms. Our variants take the Nash social welfare objective into account and select a distribution over the arms in each round instead of a single arm. For Explore-First, we derive $\widetilde{\mathcal{O}}\left(N^{2/3}K^{1/3}T^{2/3}\right)$ regret bound, which recovers the aforementioned single-agent bound with an additional factor of $N^{2/3}$. We also show that changing a parameter of the algorithm yields a regret bound of $\widetilde{\mathcal{O}}\left(N^{1/3}K^{2/3}T^{2/3}\right)$, which offers a different tradeoff between the dependence on $N$ and $K$. For Epsilon-Greedy, we recover the same two regret bounds, although the analysis becomes much more intricate. This is because, as mentioned above, Epsilon-Greedy is a horizon-independent algorithm (i.e. it does not require apriori knowledge of $T$), unlike Explore-First. For UCB, we derive $\widetilde{\mathcal{O}}\left(NKT^{1/2}\right)$ and $\widetilde{\mathcal{O}}\left(N^{1/2}K^{\frac{3}{2}}T^{1/2}\right)$ regret bounds; our dependence on $K$ worsens compared to the classical single-agent case, but importantly, we recover the same $\sqrt{T}$ dependence, which is provably optimal (see Appendix A).

Deriving these regret bounds for the multi-agent case requires overcoming two key difficulties that do not appear in the single-agent case. First, our goal is to optimize a complicated function, the Nash social welfare, rather than simply selecting the best arm. This requires a Lipschitz-continuity analysis of the Nash social welfare function and the use of new tools such as the McDiarmid's inequality which are not needed in the standard analysis. Second, the optimization is over an infinite space (the set of distributions over arms) rather than over a finite space (the set of arms). Thus, certain tricks such as a simple union bound no longer work; we use the concept of $\delta$-covering, used heavily in the Lipschitz bandit framework [20], in order to address this.

Our contributions are twofold. Conceptually, we promote a multi-agent viewpoint that requires striking a tradeoff between multiple reward functions; this can be applied to other classical single-agent problems. More technically, as explained below, the existing MAB literature does not provide a $\sqrt{T}$ bound for the Nash social welfare objective. Our regret bounds may be generalizable to a broader family of objectives for which a $\sqrt{T}$ bound was previously unknown.

## 1.2 Related Work

Since the multi-armed bandit problem was introduced by Thompson [21], many variants of it have been proposed, such as sleeping bandit [22], contextual bandit [23], dueling bandit [24], Lipschitz bandit [20], etc. However, all these variants involve optimizing the cumulative regret or identifying the Pareto frontier of multiple objectives [25] *from the perspective of a single agent*. We note that other multi-agent variants of the multi-armed bandit problem have been explored recently [26, 27], including in distributed environments [28–30]. However, they still involve a common reward like in the classical multi-armed bandit problem. Their focus is on getting the agents to cooperate to maximize this common reward.

It is possible to view our work from a single-agent prespective by treating the $K$-simplex of probability distributions over arms as a "continuum of arms" and the Nash welfare as a non-linear objective to be optimized. Kleinberg et al. [31] explore continuum-armed bandits, but establish a $\widetilde{\mathcal{O}}\left(T^{\frac{\gamma+1}{\gamma+2}}\right)$ regret bound, where $\gamma$ is the zooming dimension, which would be $\Theta(K)$ in our case. Bubeck et al. [32] also study continuum-armed bandits and establish a similar regret bound, but with respect to the near-optimality dimension, which is again $\Theta(K)$ in our case. Hence, the guarantees of Kleinberg et al. [31] and Bubeck et al. [32], when applied to our setting, are worse than our $\widetilde{\mathcal{O}}\left(\sqrt{T}\right)$ regret bound. The literature on bandit convex optimization [33–35] offers $\widetilde{\mathcal{O}}\left(\sqrt{T}\right)$ regret bound, but requires the objective to be concave, which the Nash social welfare (NSW) is not.[2]

Another key aspect of our framework is the focus on fairness. Recently, several papers have focused on fairness in the multi-armed bandit problem. For instance, Joseph et al. [37] design a UCB variant which guarantees what they refer to as meritocratic fairness to the arms, i.e., that a worse arm is never preferred to a better arm regardless of the algorithm's confidence intervals for them. Liu et al. [38] require that similar arms be treated similarly, i.e., two arms with similar mean rewards be selected with similar probabilities. Gillen et al. [39] focus on satisfying fairness with respect to an unknown fairness metric. Finally, Patil et al. [40] assume that there are external constraints requiring that

---

[2]Since NSW is log-concave, one might argue that we can apply these results to log-NSW. However, a regret bound in log-NSW does not imply the desired regret bound in NSW.

each arm be pulled in at least a certain fraction of the rounds and design algorithms that achieve low regret subject to this constraint. All these papers seek to achieve fairness *with respect to the arms*. In contrast, in our work, the arms are "inanimate" (e.g. policy decisions) and we seek fairness *with respect to the agents*, who are separate from the arms.

## 2 Preliminaries

For $n \in \mathbb{N}$, define $[n] = \{1, \ldots, n\}$. Let $N, K \in \mathbb{N}$. In the *multi-agent multi-armed bandit* (MA-MAB) problem, there is a set of *agents* $[N]$ and a set of *arms* $[K]$. For each agent $i \in [N]$ and arm $j \in [K]$, there is a *reward distribution* $D_{i,j}$ with mean $\mu^*_{i,j}$ and support $[0,1]$;[3] when arm $j$ is pulled, each agent $i$ observes an independent *reward* sampled from $D_{i,j}$. Let us refer to $\mu^* = (\mu^*_{i,j})_{i \in [N], j \in [K]} \in [0,1]^{N \times K}$ as the (true) reward matrix.

**Policies:** As mentioned in the introduction, pulling an arm deterministically may be favorable to one agent, but disastrous to another. Hence, we are interested in *probability distributions* over arms, which we refer to as *policies*. The $K$-simplex, denoted $\Delta^K$, is the set of all policies. For a policy $p \in \Delta^K$, $p_j$ denotes the probability with which arm $j$ is pulled. Note that due to linearity of expectation, the expected reward to agent $i$ under policy $p$ is $\sum_{j=1}^{K} p_j \cdot \mu^*_{i,j}$.

**Nash social welfare:** The Nash social welfare is defined the product of (expected) rewards to the agents. Given $\mu = (\mu_{i,j})_{i \in [N], j \in [K]}$, and policy $p \in \Delta^K$, define $\text{NSW}(p, \mu) = \prod_{i=1}^{N} \left( \sum_{j=1}^{K} p_j \cdot \mu_{i,j} \right)$. Thus, the (true) Nash social welfare under policy $p$ is $\text{NSW}(p, \mu^*)$. Hence, if we knew $\mu^*$, we would pick an *optimal policy* $p^* \in \arg\max_{p \in \Delta^K} \text{NSW}(p, \mu^*)$. However, because we do not know $\mu^*$ in advance, our algorithms will often produce an estimate $\widehat{\mu}$, and use it to choose a policy; the quantity $\text{NSW}(p, \widehat{\mu})$ will play a key role in our algorithms and their analysis.

**Algorithms:** An algorithm for the MA-MAB problem chooses a policy $p^t$ in each round $t \in \mathbb{N}$. Then, an arm $a^t$ is sampled according to policy $p^t$, and for each agent $i \in [N]$, a reward $X^t_{i,a^t}$ is sampled independently from distribution $D_{i,a^t}$. At the end of round $t$, the algorithm learns the sampled arm $a^t$ and the reward vector $(X^t_{i,a^t})_{i \in [N]}$, which it can use to choose policies in the later rounds.

**Reward estimates:** All our algorithms maintain an estimate of the mean reward matrix $\mu^*$ at every round. For round $t$ and arm $j \in [K]$, let $n^t_j = \sum_{s=1}^{t-1} \mathbb{1}[a^s = j]$ denote the number of times arm $j$ is pulled at the beginning of round $t$, and let $\widehat{\mu}^t_{i,j} = \frac{1}{n^t_j} \sum_{s \in [t-1]: a^s = j} X^s_{i,j}$ denote the average reward experienced by agent $i$ from the $n^t_j$ pulls of arm $j$ thus far. Our algorithms treat these as an estimate of $\mu^*_{i,j}$ available at the beginning of round $t$. Let $\widehat{\mu}^t = (\widehat{\mu}^t_{i,j})_{i \in [N], j \in [K]}$.

**Regret:** Recall that $p^*$ is an optimal policy that has the highest Nash social welfare. The *instantaneous regret* in round $t$ due to an algorithm choosing $p^t$ is $r^t = \text{NSW}(p^*, \mu^*) - \text{NSW}(p^t, \mu^*)$. The (cumulative) *regret* in round $T$ due to an algorithm choosing $p^1, \ldots, p^T$ is $R^T = \sum_{t=1}^{T} r^t$. We note that $R^T$ and $r^t$ are defined for a specific algorithm, which will be clear from the context. We are interested in bounding the *expected regret* $\mathbb{E}[R^T]$ of an algorithm at round $T$, where the expectation is over the randomness involved in sampling the arms $a^t$ and the agent rewards $(X^t_{i,a^t})_{i \in [N]}$ for $t \in [T]$.[4] We say that an algorithm is *horizon-dependent* if it needs to know $T$ in advance, and *horizon-independent* if it yields a regret bound at each possible $T$ simultaneously.

## 3 Explore-First

Perhaps the simplest algorithm (with a sublinear regret bound) in the classic single-agent MAB framework is Explore-First. It is composed of two distinct stages. The first stage is *exploration*, during which the algorithm pulls each arm $L$ times. At the end of this stage, the algorithm computes the arm $\widehat{a}$ with the best estimated mean reward, and in the subsequent *exploitation* stage, pulls arm $\widehat{a}$ in every

---

[3]We need the support of the distribution to be non-negative and bounded, but the upper bound of 1 is without loss of generality. All our bounds scale linearly with the upper bound on the support.

[4]The algorithms we study do not introduce any further randomness in choosing the policies.

round. The algorithm is horizon-dependent, i.e., it takes the horizon $T$ as input and sets $L$ as a function of $T$. Setting $L = \Theta\left(K^{-\frac{2}{3}}T^{\frac{2}{3}}\log^{\frac{1}{3}}(T)\right)$ yields regret bound $\mathbb{E}[R^T] = \mathcal{O}\left(K^{\frac{1}{3}}T^{\frac{2}{3}}\log^{\frac{1}{3}}(T)\right)$ [1].

In our multi-agent variant, presented as Algorithm 2 in the supplementary material, the exploration stage pulls each arm $L$ times as before. However, at the end of this stage, the algorithm computes, not an arm $\widehat{a}$, but a policy $\widehat{p}$ with the best estimated Nash social welfare. During exploitation, it then uses policy $\widehat{p}$ in every round. With an almost identical analysis as in the single-agent setting, we recover the aforementioned regret bound with an additional $N^{2/3}$ factor for $N$ agents.

Using a novel and more intricate argument, we show that a different tradeoff between the exponents of $N$ and $K$ can be obtained, where $N^{2/3}$ is reduced to $N^{1/3}$ at the expense of increasing $K^{1/3}$ to $K^{2/3}$ (and adding a logarithmic term). We later use this approach again in our analysis of more sophisticated algorithms.

We remark that Algorithm 2 can be implemented efficiently. The only non-trivial step is to compute the optimal policy given an estimated reward matrix, i.e., $\widehat{p} \in \arg\max_{p \in \Delta^K} \mathrm{NSW}(p, \widehat{\mu})$. Since the Nash social welfare is known to be log-concave, this can be solved in polynomial time [41].

Our analysis of Explore-First and later algorithms relies on a few elementary lemmas regarding the behavior of the Nash social welfare function $\mathrm{NSW}(p, \mu)$. We are mainly interested in how much the function can change when its arguments change. To that end, the following folklore result translates the difference in a product to a sum of point-wise differences that are easier to deal with. All missing proofs are in the supplementary material.

**Lemma 1.** *Let $a_i, b_i \in [0,1]$ for $i \in [N]$. Then, $\left|\prod_{i=1}^{N} a_i - \prod_{i=1}^{N} b_i\right| \leq \sum_{i=1}^{N} |a_i - b_i|$.*

Using Lemma 1, we can easily analyze Lipschitz-continuity of $\mathrm{NSW}(p, \mu)$ when either $p$ or $\mu$ changes and the other is fixed. First, we consider change in $p$ with $\mu$ fixed.

**Lemma 2.** *Given a reward matrix $\mu \in [0,1]^{N \times K}$ and policies $p^1, p^2 \in \Delta^K$, we have*

$$\left|\mathrm{NSW}(p^1, \mu) - \mathrm{NSW}(p^2, \mu)\right| \leq N \cdot \left\|p^1 - p^2\right\|_1 = N \cdot \sum_{j \in [K]} \left|p_j^1 - p_j^2\right|.$$

Next, we consider change in $\mu$ with $p$ fixed.

**Lemma 3.** *Given a policy $p \in \Delta^K$, and reward matrices $\mu^1, \mu^2 \in [0,1]^{N \times K}$, we have*

$$\left|\mathrm{NSW}(p, \mu^1) - \mathrm{NSW}(p, \mu^2)\right| \leq \sum_{i \in [N]} \sum_{j \in [K]} p_j \cdot \left|\mu_{i,j}^1 - \mu_{i,j}^2\right|.$$

We are now ready to establish the regret bounds for Explore-First.

**Theorem 1.** *Explore-First is horizon-dependent and has the following expected regret at round $T$.*

- *When $L = \Theta\left(N^{\frac{2}{3}}K^{-\frac{2}{3}}T^{\frac{2}{3}}\log^{\frac{1}{3}}(NKT)\right)$, $\mathbb{E}[R^T] = \mathcal{O}\left(N^{\frac{2}{3}}K^{\frac{1}{3}}T^{\frac{2}{3}}\log^{\frac{1}{3}}(NKT)\right)$.*

- *When $L = \Theta\left(N^{\frac{1}{3}}K^{-\frac{1}{3}}T^{\frac{2}{3}}\log^{\frac{2}{3}}(NKT)\right)$, $\mathbb{E}[R^T] = \mathcal{O}\left(N^{\frac{1}{3}}K^{\frac{2}{3}}T^{\frac{2}{3}}\log^{\frac{2}{3}}(NKT)\right)$.*

## 4   Epsilon-Greedy

A slightly more sophisticated algorithm than Explore-First is Epsilon-Greedy, which is presented as Algorithm 3 in the supplementary material. It spreads out exploration instead of performing it all at the beginning. Specifically, at each round $t$, it performs exploration with probability $\epsilon^t$, and exploitation otherwise. Exploration cycles through the arms in a round-robin fashion, while exploitation uses the policy $p^t$ with the highest Nash social welfare under the current estimated reward matrix (rather than choosing a single estimated best arm as in the classical algorithm).

Like Explore-First, Epsilon-Greedy can be implemented efficiently. The only non-trivial step is to compute $\widehat{p} \in \arg\max_{p \in \Delta^K} \mathrm{NSW}(p, \widehat{\mu})$, which, as mentioned before, can be done efficiently.

The key advantage of Epsilon-Greedy over Explore-First is that it is horizon-independent. However, in the $\widehat{\mu}$ computed in Explore-First at the end of exploration, each $\widehat{\mu}_{i,j}$ is the average of $L$ iid samples, where $L$ is fixed. In contrast, in the $\widehat{\mu}^t$ computed in Epsilon-Greedy in round $t$, each $\widehat{\mu}_{i,j}^t$ is the

average of $n_j^t$ iid samples. The fact that $n_j^t$ is itself a random variable and the $\widehat{\mu}_{i,j}^t$-s are correlated through the $n_j^t$-s prevents a direct application of certain statistical inequalities, thus complicating the analysis of Epsilon-Greedy. To address this, we first present a sequence of useful lemmas that apply to *any* algorithm, and then use them to prove the regret bounds of Epsilon-Greedy and later UCB.

## 4.1 Useful Lemmas

Recall that $\mu^*$ and $\widehat{\mu}^t$ denote the true reward matrix and the estimated reward matrix at the beginning of round $t$, respectively. Our goal is to find an upper bound on the quantity $|\mathrm{NSW}(p, \mu^*) - \mathrm{NSW}(p, \widehat{\mu}^t)|$ that, with high probability, holds at every $p \in \Delta^K$ simultaneously. To that end, we first need to show that $\widehat{\mu}^t$ will be close to $\mu^*$ with high probability.

Recall that random variable $n_j^t$ denotes the number of times arm $j$ is pulled by an algorithm before round $t$, and $\widehat{\mu}_{i,j}^t$ is an average over $n_j^t$ independent samples. Hence, we cannot directly apply Hoeffding's inequality, but we can nonetheless use standard tricks from the literature.

**Lemma 4.** *Define $r_j^t = \sqrt{\frac{2 \log(NKt)}{n_j^t}}$, and event $\mathcal{E}^t \triangleq \forall i \in [N], j \in [K] : \left| \widehat{\mu}_{i,j}^t - \mu_{i,j}^* \right| \leq r_j^t$. Then, for any algorithm and any $t$, we have $\Pr[\mathcal{E}^t] \geq 1 - \frac{2}{t^3}$.*

Conditioned on $\mathcal{E}^t$, we wish to bound $|\mathrm{NSW}(p, \mu^*) - \mathrm{NSW}(p, \widehat{\mu}^t)|$ simultaneously at all $p \in \Delta^K$. We provide two such (incomparable) bounds, which will form the crux of our regret bound analysis. The first bound is a direct application of the Lipschitz-continuity analysis from Lemma 3.

**Lemma 5.** *Conditioned on $\mathcal{E}^t$, we have that*
$$\forall p \in \Delta^K : \left| \mathrm{NSW}(p, \widehat{\mu}^t) - \mathrm{NSW}(p, \mu^*) \right| \leq N \cdot \sum_{j \in [K]} p_j \cdot r_j^t.$$

The factor of $N$ in Lemma 5 stems from analyzing how much $\widehat{\mu}^t$ may deviate from $\mu^*$ conditioned on $\mathcal{E}^t$, in the worst case. However, even after conditioning on $\mathcal{E}^t$, $\widehat{\mu}^t$ remains a random variable. Hence, one may expect that its deviation, and thus the difference $|\mathrm{NSW}(p, \widehat{\mu}^t) - \mathrm{NSW}(p, \mu^*)|$, may be smaller in expectation. Thus, to derive a different bound than in Lemma 5, we wish to apply McDiarmid's inequality. However, there are two issues in doing so directly.

- McDiarmid's inequality bounds the deviation of $\mathrm{NSW}(p, \widehat{\mu}^t)$ from its expected value. If $\widehat{\mu}^t$ consisted of independent random variables, like in Explore-First, this would be equal to $\mathrm{NSW}(p, \mu^*)$. However, in general, these variables may be correlated through $n_j^t$. We use a conditioning trick to address this issue.

- We cannot hope to apply McDiarmid's inequality at each $p \in \Delta^K$ separately and use the union bound because $\Delta^K$ is infinite. So we apply it at each $p$ in a $\delta$-cover of $\Delta^K$, apply the union bound, and then translate the guarantee to nearby $p \in \Delta^K$ using the Lipschitz-continuity analysis from Lemma 2.

The next result is one of the key technical contributions of our work with a rather long proof.

**Lemma 6.** *Define the event*
$$\mathcal{H}^t \triangleq \forall p \in \Delta^K : \left| \mathrm{NSW}(p, \widehat{\mu}^t) - \mathrm{NSW}(p, \mu^*) \right| \leq \sqrt{12NK \log(NKt)} \cdot \sum_{j \in [K]} p_j \cdot r_j^t + \frac{4}{t}.$$

*Then, for any algorithm and any $t$, we have $\Pr[\mathcal{H}^t | \mathcal{E}^t] \geq 1 - 2/t^3$.*

Finally, we use the following simple observation in deriving our asymptotic bounds.

**Proposition 1.** *For constant $p \in \mathbb{R}$, $\sum_{t=1}^{T} t^p$ is $\Theta(\log T)$ if $p = -1$ and $\Theta(T^{p+1})$ when $p > -1$.*

## 4.2 Analysis of Epsilon-Greedy

We can now use these lemmas to establish the regret bounds for Epsilon-Greedy.

**Theorem 2.** *Epsilon-Greedy is horizon-independent, and has the following expected regret at any round $T$.*

- *If $\epsilon^t = \Theta\left(N^{\frac{2}{3}} K^{\frac{1}{3}} t^{-\frac{1}{3}} \log^{\frac{1}{3}}(NKt)\right)$ for all $t$, $\mathbb{E}[R^T] = \mathcal{O}\left(N^{\frac{2}{3}} K^{\frac{1}{3}} T^{\frac{2}{3}} \log^{\frac{1}{3}}(NKT)\right)$.*

- *If $\epsilon^t = \Theta\left(N^{\frac{1}{3}} K^{\frac{2}{3}} t^{-\frac{1}{3}} \log^{\frac{2}{3}}(NKt)\right)$ for all $t$, $\mathbb{E}[R^T] = \mathcal{O}\left(N^{\frac{1}{3}} K^{\frac{2}{3}} T^{\frac{2}{3}} \log^{\frac{2}{3}}(NKT)\right)$.*

# 5 UCB

---

**Algorithm 1:** UCB

---

**Input:** Number of agents $N$, number of arms $K$

**Parameters :** Confidence parameter $\alpha^t$ for each $t \in \mathbb{N}$

`// Pull each arm once`

**for** $t = 1, \ldots, K$ **do**

$\quad\mid\quad$ $p^t \leftarrow$ policy that puts probability 1 on arm $t$ $\qquad$ `// Pull arm `$t$` deterministically`

**end**

**for** $t = K + 1, \ldots$ **do**

$\quad\mid\quad$ Compute the estimated reward matrix $\widehat{\mu}^t$

$\quad\mid\quad$ $p^t \leftarrow \arg\max_{p \in \Delta^K} \text{NSW}(p, \widehat{\mu}^t) + \alpha^t \sum_{j \in [K]} p_j \cdot r_j^t$, where $r_j^t \triangleq \sqrt{\frac{\log(NKt)}{n_j^t}}$.

**end**

---

In the classical multi-armed bandit setting, UCB first pulls each arm once. Afterwards, it merges exploration and exploitation cleverly by pulling, in each round, an arm maximizing the sum of its estimated reward and a confidence interval term similar to $r_j^t$ in Algorithm 1. Our multi-agent variant similarly selects a policy that maximizes the estimated Nash social welfare plus a confidence term for a policy, which simply takes a linear combination of the confidence intervals of the arms.

Unlike Explore-First and Epsilon-Greedy, it is not clear if our UCB variant admits an efficient implementation due to the step of computing $\arg\max_{p \in \Delta^K} \text{NSW}(p, \widehat{\mu}) + \alpha^t \sum_{j \in [K]} p_j r_j^t$. Due to the added linear term, the objective is no longer log-concave. This remains a challenging open problem. However, we notice that this can also be viewed as the problem of optimizing a polynomial over a simplex, which, while NP-hard in general, admits a PTAS when the degree is a constant [42, 43]. Hence, when the number of agents $N$ is a constant, this step can be computed approximately, but it remains to be seen how this approximation translates to the regret bound.

We show that UCB achieves the desired $\sqrt{T}$ dependence on the horizon (up to logarithmic factors). In Appendix A, we show that this is optimal.

**Theorem 3.** *UCB is horizon-independent, and has the following expected regret at any round $T$.*

- *If $\alpha^t = N$ for all $t$, $\mathbb{E}[R^T] = \mathcal{O}\left(NKT^{\frac{1}{2}} \log(NKT)\right)$.*

- *If $\alpha^t = \sqrt{12NK \log(NKt)}$ for all $t$, $\mathbb{E}[R^T] = \mathcal{O}\left(N^{\frac{1}{2}} K^{\frac{3}{2}} T^{\frac{1}{2}} \log^{\frac{3}{2}}(NKT)\right)$.*

*Proof.* Fix one of two parameter choices:

1. $\alpha^t = N$ for all $t$ and $c = N$.
2. $\alpha^t = \sqrt{12NK \log(NKt)}$ for all $t$ and $c = \sqrt{12NK \log(NKT)}$.

Note that in both cases, $\alpha^t \leq c$ for all $t$. Hence, $c$ serves as an upper bound on $\alpha^t$ that does not depend on $t$. We show that in both cases, running UCB with the $\alpha^t$ parameter value yields a regret bound of $\mathbb{E}[R^T] = O(cK\sqrt{T} \log(NKT))$. Substituting the two choices of $c$ then yields the two regret bounds. Let us again focus on the event

$$\mathcal{C}_\alpha^t \triangleq \forall p \in \Delta^K : \left|\text{NSW}(p, \mu^*) - \text{NSW}(p, \widehat{\mu}^t)\right| \leq \alpha^t \cdot \sum_{j \in [K]} p_j \cdot r_j^t + \frac{4}{t}.$$

Recall the clean events $\mathcal{E}^t$ and $\mathcal{H}^t$ defined in Lemmas 4 and 6. Conditioned on $\mathcal{E}^t \wedge \mathcal{H}^t$, note that $\mathcal{C}_\alpha^t$ holds for $\alpha^t = N$ due to Lemma 5, and for $\alpha^t = \sqrt{12NK \log NKt}$ due to Lemma 6. Using Lemmas 4 and 6, and the union bound, we have $\Pr[\neg \mathcal{C}_\alpha^t] \leq 1 - \Pr[\mathcal{E}^t \wedge \mathcal{H}^t] = 1 - \Pr[\mathcal{E}^t] \cdot \Pr[\mathcal{H}^t | \mathcal{E}^t] \leq 1 - (1 - 2/t^3) \cdot (1 - 2/t^3) \leq 4/t^3$.

Define a clean event $\mathcal{C}_\alpha^* \triangleq \bigwedge_{t \geq \sqrt{T}} \mathcal{C}_\alpha^t$. Here, we do not care about the first $\sqrt{T}$ rounds because the maximum regret from these rounds is $\mathcal{O}\left(\sqrt{T}\right)$, which is permissible given our desired regret bounds. By the union bound, we have $\Pr[\neg \mathcal{C}_\alpha^*] \leq T \cdot 4/(\sqrt{T})^3 = 4/\sqrt{T}$. Thus, $\mathcal{C}_\alpha^*$ is a high-probability event. In what follows, we derive an upper bound on the expected regret conditioned on $\mathcal{C}_\alpha^*$, i.e., $\mathbb{E}[R^T|\mathcal{C}_\alpha^*]$. Since conditioning on a high-probability event does not affect the expected value significantly, the desired regret bound will then follow. For any $t \in [T]$, conditioned on $\mathcal{C}_\alpha^t$ we have that

$$\text{NSW}(p^*, \mu^*) \leq \text{NSW}(p^*, \widehat{\mu}^t) + \alpha^t \sum_{j \in [K]} p_j^* \cdot r_j^t + \frac{4}{t} \leq \text{NSW}(p^t, \widehat{\mu}^t) + \alpha^t \sum_{j \in [K]} p_j^t \cdot r_j^t + \frac{4}{t}$$

$$\leq \text{NSW}(p^t, \mu^*) + 2\alpha^t \sum_{j \in [K]} p_j^t \cdot r_j^t + \frac{8}{t},$$

where the first and the last transition are from conditioning on $\mathcal{C}_\alpha^t$, and the second transition is because $p = p^t$ maximizes the quantity $\text{NSW}(p, \widehat{\mu}^t) + \alpha^t \sum_{j \in [K]} p_j \cdot r_j^t$ in the UCB algorithm.

Let us write $p^{[T]} = (p^1, \ldots, p^T)$ for the random variable denoting the policies used by the algorithm, and $\overline{p}^{[T]} = (\overline{p}^1, \ldots, \overline{p}^T)$ to denote a specific value in $(\Delta^K)^T$ taken by the random variable.

Instead of analyzing $\mathbb{E}[R^T|\mathcal{C}_\alpha^*]$ directly, we further condition on UCB choosing a specific sequence of policies $\overline{p}^{[T]}$. That is, we are interested in deriving an upper bound on $\mathbb{E}[R^T|\mathcal{C}_\alpha^* \wedge p^{[T]} = \overline{p}^{[T]}]$.[5] Interestingly, we show that this quantity is $\mathcal{O}\left(cK\sqrt{T}\log(NKT)\right)$ for *every* possible $\overline{p}^{[T]}$.

Fix an arbitrary $\overline{p}^{[T]}$. For $t \in [T]$ and $j \in [K]$, define $q_j^t = \sum_{s=1}^t \overline{p}_j^s$. Then, $\mathbb{E}[n_j^t|p^{[T]} = \overline{p}^{[T]}] = q_j^t$. For each $j \in [K]$, let $T_j$ be the smallest $t$ for which $q_j^t \geq 2\sqrt{T \log(NKT)}$ (if no such $t$ exists, let $T_j = T$); note that given $\overline{p}^{[T]}$, $T_j$ is fixed and not a random variable. Also, we have that $q_j^{T_j} = \Theta\left(\sqrt{T \log(NKT)}\right)$ for each $j \in [K]$.

Let us define a clean event $\mathcal{B} \triangleq \forall j \in [K], n_j^{T_j} \geq \sqrt{T \log(NKT)}$. We first show that this is a high probability event. Indeed, using Hoeffding's inequality, we have that for each $j \in [K]$,

$$\Pr\left[n_j^{T_j} < \sqrt{T \log(NKT)} \,\Big|\, \mathcal{C}_\alpha^* \wedge p^{[T]} = \overline{p}^{[T]}\right] \leq \Pr\left[n_j^{T_j} < s_j^{T_j} - \sqrt{T \log(NKT)} \,\Big|\, \mathcal{C}_\alpha^* \wedge p^{[T]} = \overline{p}^{[T]}\right]$$

$$\leq \frac{1}{N^2 K^2 T^2}.$$

Taking union bound over $j \in [K]$, we have that $\Pr\left[\neg \mathcal{B} \,\big|\, \mathcal{C}_\alpha^* \wedge p^{[T]} = \overline{p}^{[T]}\right] \leq \frac{1}{N^2 K T^2}$.

Next, we bound $\mathbb{E}[R^T|\mathcal{C}_\alpha^* \wedge p^{[T]} = \overline{p}^{[T]}]$ by using event $\mathcal{B}$.

$$\mathbb{E}\left[R^T \,\big|\, \mathcal{C}_\alpha^* \wedge p^{[T]} = \overline{p}^{[T]}\right] = \sum_{t=1}^T \mathbb{E}\left[\text{NSW}(p^*, \mu^*) - \text{NSW}(p^t, \mu^*) \,\Big|\, \mathcal{C}_\alpha^* \wedge p^{[T]} = \overline{p}^{[T]}\right]$$

$$\leq \max(K, \sqrt{T}) + \sum_{t=\max(K,\sqrt{T})+1}^T \left(1 \cdot \mathbb{E}\left[\text{NSW}(p^*, \mu^*) - \text{NSW}(p^t, \mu^*) \,\big|\, \mathcal{C}_\alpha^* \wedge p^{[T]} = \overline{p}^{[T]} \wedge \mathcal{B}\right]\right.$$

$$\left. + \Pr\left[\neg \mathcal{B} \,\big|\, \mathcal{C}_\alpha^* \wedge p^{[T]} = \overline{p}^{[T]}\right] \cdot 1\right)$$

$$\leq \max(K, \sqrt{T}) + \sum_{t=\max(K,\sqrt{T})+1}^T \mathbb{E}\left[2\alpha^t \sum_{j \in [K]} \overline{p}_j^t \cdot r_j^t + \frac{8}{t} \,\bigg|\, \mathcal{C}_\alpha^* \wedge p^{[T]} = \overline{p}^{[T]} \wedge \mathcal{B}\right]$$

$$+ T \cdot \Pr\left[\neg \mathcal{B} \,\big|\, \mathcal{C}_\alpha^* \wedge p^{[T]} = \overline{p}^{[T]}\right]$$

$$\leq \max(K, \sqrt{T}) + 1 + 2c\sqrt{2\log(NKT)} \sum_{t=1}^T \sum_{j \in [K]} \frac{\overline{p}_j^t}{\sqrt{c_j}}. \tag{1}$$

---

[5]Note that even after conditioning on $p^{[T]} = \overline{p}^{[T]}$, there is still randomness left in sampling actions from the policies and sampling the rewards of those actions.

The final transition holds because $\alpha^t \leq c$ for all $t$, $r_j^t = \sqrt{\frac{2\log(NKT)}{n_j^t}}$, and conditioned on $\mathcal{B}$, $n_j^t \geq c_j$ for each $j \in [K]$ and $t \in [T]$, where $c_j = 1$ if $t < T_j$, and $c_j = \sqrt{T\log(NKT)}$ if $t \geq T_j$. Hence,

$$
\mathbb{E}\left[ R^T \,\middle|\, \mathcal{C}_\alpha^* \wedge p^{[T]} = \overline{p}^{[T]} \right] \leq \max(K, \sqrt{T}) + 1 + 2c\sqrt{2\log(NKT)} \sum_{j \in [K]} \sum_{t=1}^{T} \frac{\overline{p}_j^t}{\sqrt{c_j}}
$$

$$
= \max(K, \sqrt{T}) + 1 + 2c\sqrt{2\log(NKT)} \sum_{j \in [K]} \left( \sum_{t=1}^{T_j - 1} \frac{\overline{p}_j^t}{1} + \sum_{t=T_j}^{T} \frac{\overline{p}_j^t}{\sqrt{T\log(NKT)}} \right)
$$

$$
\leq \max(K, \sqrt{T}) + 1 + 2c\sqrt{2\log(NKT)} \sum_{j \in [K]} \left( q_j^{T_j} + \frac{T}{\sqrt{T\log(NKT)}} \right) = \mathcal{O}\left( cK\sqrt{T}\log(NKT) \right).
$$

Because this bound holds for every possible $\overline{p}^{[T]}$, we also have that $\mathbb{E}[R^T | \mathcal{C}_\alpha^*] = \mathcal{O}\left( cK\sqrt{T}\log(NKT) \right)$. Finally, we can see that

$$
\mathbb{E}[R^T] = \Pr[\mathcal{C}_\alpha^*] \cdot \mathbb{E}[R^T | \mathcal{C}_\alpha^*] + \Pr[\neg \mathcal{C}_\alpha^*] \cdot \mathbb{E}[R^T | \neg \mathcal{C}_\alpha^*]
$$

$$
\leq 1 \cdot \mathcal{O}\left( cK\sqrt{T}\log(NKT) \right) + \frac{4}{\sqrt{T}} \cdot 1 = \mathcal{O}\left( cK\sqrt{T}\log(NKT) \right).
$$

Recall that substituting $c = N$ and $c = \sqrt{12NK\log(NKT)}$ yields the two regret bounds. $\qquad\square$

We emphasize that our analysis of the multi-agent UCB differs significantly from the analysis of the classical (single-agent) UCB. For example, the use of clean event $\mathcal{C}_\alpha^*$ is unique to our analysis. More importantly, the expression in Equation (1) is also unique to our setting in which the algorithm can "pull" a probability distribution over the arms. The corresponding expression in case of the classical UCB turns out to be much simpler and straightforward to bound. In contrast, we need to use additional tricks to derive the bound of $\mathcal{O}\left( cK\sqrt{T}\log(NKT) \right)$.

Finally, in the proof presented above, we showed that, assuming the clean event $\mathcal{C}_\alpha^*$, the expected regret is small conditioned on *any* sequence of policies that the UCB algorithm might use. At the first glance, this may seem surprising. However, a keen reader can observe that the clean event $\mathcal{C}_\alpha^*$ can only occur when the UCB algorithm uses a "good" sequence of policies that leads to low expected regret. A similar phenomenon is observed in the analysis of the classical (single-agent) UCB algorithm as well (see, e.g., [1]): assuming a different clean event, the classical UCB algorithm is guaranteed to not pull suboptimal arms too many times.

## 6   Discussion

Our work leaves open several directions for the future. There are immediate technical challenges such as a polynomial-time implementation and a logarithmic instance-dependent regret bound for our UCB variant (see Section 5), and deriving improved lower bounds which scale with the number of agents $N$ (see Appendix A). It would also be interesting to consider other fairness notions such as the egalitarian welfare or the core [4]. More broadly, our work promotes a multi-agent variant in which one must strike a tradeoff between the reward functions of multiple agents. An exciting future direction would be to consider such multi-agent variants of other classical single-agent decision-making problems.

## Acknowledgments and Disclosure of Funding

Nisarg Shah acknowledges support from an NSERC Disocvery Grant.

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
