# Appendix

## A  Lower Bound

In this section, we establish a lower bound on the expected regret of any algorithm for our multi-agent multi-armed bandit (MA-MAB) problem. In the classical multi-armed bandit problem, it is known that no algorithm can achieve a regret bound of $E[R^T] = o(\sqrt{KT})$, when the constant inside the little-Oh notation is required to be independent of the distributions in the given instance [18]. Our goal in this section is twofold. First, we argue that, due to subtle reasons, a lower bound for the classical multi-armed bandit problem does not immediately carry over to our setting. Nevertheless, we subsequently argue that a very simple adjustment to the proof of the classical lower bound due to Auer et al. [18] is sufficient to extend it to our problem. To avoid excessive repetition of notation and proof arguments, we purposefully leave this section *not* self-contained and only outline these adjustments needed. We refer an interested reader to the work of Auer et al. [18] for the full and detailed proof. That said, this argument establishes that the $\sqrt{T}$-dependence of the expected regret of our UCB variant from Theorem 3 is optimal. Note that our focus is solely on the dependence of the expected regret on $T$ as $T$ is typically much larger than both the number of agents $N$ and the number of arms $K$. We leave it to future work to optimize the dependence on $N$ and $K$.

First, we notice that any lower bound derived for the case of a single agent also holds when there are $N > 1$ agents. This is because one can consider instances in which all but one of the agents derive a fixed reward of 1 from every arm. Note that the contribution of such agents to the product in the Nash social welfare expression is always 1 regardless of the policy chosen. Hence, the Nash social welfare reduces to simply the expected utility of the remaining agent, i.e., the Nash social welfare in an instance with only this one agent. Therefore, any lower bound on the expected regret that holds for MA-MAB with a single agent also holds for MA-MAB with $N > 1$ agents.

Next, let us focus on the MA-MAB problem with $N = 1$ agent. At the first glance, this may look almost identical to the classical multi-armed bandit problem. After all, if there is but one agent, the policy maximizing the Nash social welfare places probability 1 on the arm $j^*$ that gives the highest mean reward to the agent. Thus, like in the classical problem, our goal would be to converge to pulling arm $j^*$ repeatedly and our regret would also be measured with respect to the best policy which deterministically pulls arm $j^*$ in every round. However, there are two subtle differences which prevent us from directly borrowing the classical lower bound.

1. In our MA-MAB problem, an algorithm is allowed to "pull" a *distribution over the arms $p^t$* in round $t$ and learn the stochastically generated rewards for a *random* arm $j^t$ sampled from this distribution. This makes the algorithm slightly more powerful than an algorithm in the classical MAB problem which must deterministically choose an arm to pull.

2. In our MA-MAB problem, the regret in round $t$ is computed as the difference between the mean reward of the best arm and the *expected* mean reward of an arm $j^t$ sampled according to the distribution $p^t$ used by the algorithm. In the classical problem, one would replace the latter term with the mean reward of the arm actually pulled in round $t$.

The latter distinction is not particularly troublesome because our focus is on the *expected* regret of an algorithm anyway. However, the first distinction makes it impossible to directly borrow lower bounds from the classical MAB problem.

One might wonder if there is still a way to reduce the MA-MAB problem with $N = 1$ agent to the classical MAB problem. For example, given an algorithm $A$ for MA-MAB with $N = 1$, what if we construct an algorithm $\widehat{A}$ for the classical MAB and use the lower bound on the expected regret of $\widehat{A}$ to derive a lower bound on the expected regret of $A$? The problem with such reduction is that once $A$ chooses a distribution $p^t$, we have no control over which arm will be sampled. This choice is crucial as it will determine what information the algorithm gets to learn. We cannot mimic this learning process in our deterministic algorithm $\widehat{A}$. Upon careful consideration, it also seems difficult to express the expected regret of $A$ as the convex combination of the expected regret of several deterministic algorithms for the classical MAB.

Instead of aiming to find a black-box reduction to the classical problem, we therefore investigate in detail the proof of the $\Omega\left(\sqrt{KT}\right)$ lower bound for the classical MAB due to Auer et al. [18,

Theorem 5.1] and observe that their argument goes through for our MA-MAB problem as well. Instead of repeating their proof, we survey the key steps of their proof in which they assume the algorithm to be deterministically pulling an arm and highlight why the argument holds even when this is not the case.

- In the proof of their Lemma A.1, in the explanation of their Equation (30), they cite the assumption that given the rewards observed in the first $t-1$ rounds (they denote this by the vector $\mathbf{r}^{t-1}$), the algorithm pulls a fixed arm $i_t$ in round $t$. They refer to the distribution $\mathbf{P}_i\{r^t|\mathbf{r}^{t-1}\}$ of the reward in round $t$ given $\mathbf{r}^{t-1}$. In their case, the randomness in $r^t$ is solely due to stochasticity of the rewards since the arm pulled $(i_t)$ is fixed. However, in our case, one can think of $\mathbf{P}_i\{r^t|\mathbf{r}^{t-1}\}$ as containing randomness both due to the random choice of $i_t$ and due to the stochasticity of the rewards, and their equations still go through.

- In the same equation, they consider $\mathbf{P}_{unif}\{i_t = i\}$, the probability that arm $i$ is pulled in round $t$. In their case, the only randomness is due to $\mathbf{r}^{t-1}$. In our case, there is additional randomness due to the sampling of an arm in round $t$ from a distribution $p^t$. However, this does not affect their calculations.

- Finally, in the proof of their Theorem A.2, they again consider the probability $\mathbf{P}_i\{i_t = i\}$ and the same argument as above ensures that their proof continues to hold in our setting.

Thus, we have the following lower bound.

**Proposition 2.** *For any algorithm for the MA-MAB problem, there exists a problem instance such that* $\mathbb{E}[R^T] = \Omega\left(\sqrt{KT}\right)$.

# B   Missing Algorithms & Proofs

---
**Algorithm 2:** Explore-First

---
**Input:** Number of agents $N$, number of arms $K$, horizon $T$
**Parameters :** Exploration period $L$
// Pull each arm $L$ times
**for** $t = 1, \ldots, K \cdot L$ **do**                                      // Exploration
   | $j \leftarrow \lceil t/L \rceil$
   | $p^t \leftarrow$ policy that puts probability 1 on arm $j$       // Pull arm $j$ deterministically
**end**
Compute the estimated reward matrix $\widehat{\mu} \triangleq \widehat{\mu}^{K\cdot L+1}$ of the rewards observed so far
Compute $\widehat{p} \in \arg\max_{p\in\Delta^K} \mathrm{NSW}(p, \widehat{\mu})$
**for** $t = K \cdot L + 1, \ldots, T$ **do**                                      // Exploitation
   | $p^t \leftarrow \widehat{p}$
**end**

---

$\delta$**-Covering:** Given a metric space $(X, d)$ and $\delta > 0$, a set $S \subseteq X$ is called a $\delta$-*cover* if for each $x \in X$, there exists $s \in S$ with $d(x, s) \leq \delta$. That is, from each point in the metric space, there is a point in the $\delta$-cover that is no more than $\delta$ distance away. We will heavily use the fact that there exists a $\delta$-cover of $(\Delta^K, \|\cdot\|_1)$ (i.e. the $K$-simplex under the $L_1$ distance) with size at most $(1 + 2/\delta)^K$ [44, p. 126], which follows from a simple discretization of the simplex.

**Algorithm 3:** $\epsilon^t$-Greedy

---

**Input:** Number of agents $N$, number of arms $K$

**Parameters:** Exploration probabilities $\epsilon^t$ for $t \in \mathbb{N}$

$curr \leftarrow 1$                  // Next arm to pull during exploration

**for** $t = 1, 2, \ldots,$ **do**

     Toss a coin with success probability $\epsilon^t$

     **if** *success* **then**                  // Exploration

         // Round-robin among arms during exploration

         $p^t \leftarrow$ policy that puts probability 1 on arm $curr$    // Pull it deterministically

         $curr \leftarrow curr + 1$                 // When $curr$ becomes $K+1$, reset to 1

     **else**                             // Exploitation

         Compute the estimated reward matrix $\widehat{\mu}^t$ from the rewards observed so far

         $p^t \leftarrow \arg\max_{p \in \Delta^K} \text{NSW}(p, \widehat{\mu}^t)$

     **end**

**end**

---

## B.1 Proof of Lemma 1

*Proof.* We prove this using induction on $N$. For $N = 1$, the lemma trivially holds. Suppose it holds for $N = n$. For $N = n + 1$, we have

$$\left| \prod_{i=1}^{n+1} a_i - \prod_{i=1}^{n+1} b_i \right| = \left| \prod_{i=1}^{n+1} a_i - b_{n+1} \prod_{i=1}^{n} a_i + b_{n+1} \prod_{i=1}^{n} a_i - \prod_{i=1}^{n+1} b_i \right|$$

$$\leq \left( \prod_{i=1}^{n} a_i \right) |a_{n+1} - b_{n+1}| + b_{n+1} \cdot \left| \prod_{i=1}^{n} a_i - \prod_{i=1}^{n} b_i \right|$$

$$\leq |a_{n+1} - b_{n+1}| + \sum_{i=1}^{n} |a_i - b_i| = \sum_{i=1}^{n+1} |a_i - b_i|,$$

where the second transition is due to the triangle inequality, and the third transition holds due to the induction hypothesis and because $a_i, b_i \in [0, 1]$ for each $i$. $\qquad\square$

## B.2 Proof of Lemma 2

*Proof.* Using Lemma 1, we have

$$\left| \text{NSW}(p^1, \mu) - \text{NSW}(p^2, \mu) \right| \leq \sum_{i \in [N]} \left| \sum_{j \in [K]} (p_j^1 - p_j^2) \cdot \mu_{i,j} \right| \leq N \cdot \sum_{j \in [K]} \left| p_j^1 - p_j^2 \right|,$$

where the final transition is due to the triangle inequality and because $\mu_{i,j} \in [0, 1]$ for each $i, j$. $\quad\square$

## B.3 Proof of Lemma 3

*Proof.* Again, using Lemma 1, we have

$$\left| \text{NSW}(p, \mu^1) - \text{NSW}(p, \mu^2) \right|$$

$$\leq \sum_{i \in [N]} \left| \sum_{j \in [K]} p_j \cdot (\mu_{i,j}^1 - \mu_{i,j}^2) \right| \leq \sum_{i \in [N], j \in [K]} p_j \cdot \left| \mu_{i,j}^1 - \mu_{i,j}^2 \right|,$$

where the last transition is due to the triangle inequality. $\qquad\square$

### B.4 Proof of Theorem 1

*Proof.* Note that the instantaneous regret $r^t(p^t)$ in any round $t$ can be at most 1 because $\text{NSW}(p, \mu^*) \in [0, 1]$ for every policy $p$. Thus,

$$\mathbb{E}[R^T] = \sum_{t=1}^{T} \mathbb{E}[r^t] \leq KL \cdot 1 + (T - KL) \cdot \mathbb{E}[\text{NSW}(p^*, \mu^*) - \text{NSW}(\widehat{p}, \mu^*)]. \tag{2}$$

Thus, our goal is to bound $\mathbb{E}[\text{NSW}(p^*, \mu^*) - \text{NSW}(\widehat{p}, \mu^*)]$. We bound this in two ways.

**First approach:** We present this approach briefly since it largely mimics the classical analysis with an application of Lemma 3. Here, we bound how much $\widehat{\mu}$ can deviate from $\mu^*$. Specifically, we let $\epsilon = \sqrt{\frac{\log(NKT)}{L}}$ and define the event $\mathcal{E} \triangleq \forall i \in [N], \forall j \in [K] : \left|\widehat{\mu}_{i,j} - \mu^*_{i,j}\right| \leq \epsilon$. Since $L$ is fixed, we have $\mathbb{E}[\widehat{\mu}_{i,j}] = \mu^*_{i,j}$. Hence, we can apply Hoeffding's inequality followed by the union bound to derive $\Pr[\mathcal{E}] \geq 1 - 2/T^2$. Conditioned on $\mathcal{E}$, from Lemma 3 we have $\text{NSW}(p, \mu^*) - \text{NSW}(p, \widehat{\mu}) \leq N\epsilon$ for every policy $p$, which implies

$$\text{NSW}(p^*, \mu^*) \leq \text{NSW}(p^*, \widehat{\mu}) + N\epsilon \leq \text{NSW}(\widehat{p}, \widehat{\mu}) + N\epsilon \leq \text{NSW}(\widehat{p}, \mu^*) + 2N\epsilon,$$

where the second transition is because $\widehat{p} \in \arg\max_{p \in \Delta^K} \text{NSW}(p, \widehat{\mu})$. Substituting this into Equation (2), using the fact that $\mathbb{E}[\text{NSW}(p^*, \mu^*) - \text{NSW}(\widehat{p}, \mu^*)] \leq 1 \cdot \mathbb{E}[\text{NSW}(p^*, \mu^*) - \text{NSW}(\widehat{p}, \mu^*)|\mathcal{E}] + \Pr[\neg\mathcal{E}] \cdot 1$, and setting $L = \Theta\left(N^{\frac{2}{3}} K^{-\frac{2}{3}} T^{\frac{2}{3}} \log^{\frac{1}{3}}(NKT)\right)$ yields the first regret bound.

**Second approach:** We now focus on another approach for bounding $\mathbb{E}[\text{NSW}(p^*, \mu^*) - \text{NSW}(\widehat{p}, \mu^*)]$, which is more intricate and offers a different tradeoff between the dependence on $N$ and $K$. Notice that for a given $p$, $\mathbb{E}[\text{NSW}(p, \widehat{\mu})] = \text{NSW}(p, \mu^*)$ because all $\widehat{\mu}_{i,j}$-s are independent and expectation decomposes over sums and products of independent random variables. Thus, we can use McDiarmid's inequality to bound $|\text{NSW}(p, \widehat{\mu}) - \text{NSW}(p, \mu^*)|$ at a given $p$.

Fix a $\delta$-cover $\mathcal{P}$ of $(\Delta^K, \|\cdot\|_1)$ with $|\mathcal{P}| \leq (1 + 2/\delta)^K$. Fix $p \in \mathcal{P}$. Notice that $\widehat{\mu}_{i,j} = (1/L) \cdot \sum_{s=1}^{L} X_{i,j}^s$, where $X_{i,j}^s$ is the reward to agent $i$ from the $s$-th pull of arm $j$ during the exploration phase.

We thus decompose $\widehat{\mu}$ into $N \cdot L$ random variables: for each $i \in [N]$ and $s \in [L]$, we let $X_i^s = (X_{i,j}^s)_{j \in [K]}$. To apply McDiarmid's inequality, we need to analyze the maximum amount $c_i^s$ by which changing $X_i^s$ can change $\text{NSW}(p, \widehat{\mu})$. Using Lemma 3, it is easy to see that $c_i^s \leq 1/L$ for each $i \in [N]$ and $s \in [L]$. Now, applying McDiarmid's inequality, we have

$$\Pr\left[|\text{NSW}(p, \widehat{\mu}) - \text{NSW}(p, \mu^*)| \leq \epsilon\right] \leq 2e^{\frac{-2\epsilon^2}{\sum_{i \in [N], s \in [L]} (c_i^s)^2}} = 2e^{\frac{-2L\epsilon^2}{N}}.$$

Setting $\epsilon = \sqrt{\frac{N \log(|\mathcal{P}|T)}{2L}}$, we have that for each $p \in \mathcal{P}$,

$$\Pr\left[|\text{NSW}(p, \widehat{\mu}) - \text{NSW}(p, \mu^*)| \leq \sqrt{\frac{N \log(|\mathcal{P}|T)}{2L}}\right] \leq \frac{2}{|\mathcal{P}|T}.$$

Using the union bound, we have that

$$\Pr\left[\forall p \in \mathcal{P} : |\text{NSW}(p, \widehat{\mu}) - \text{NSW}(p, \mu^*)| \leq \sqrt{\frac{N \log(|\mathcal{P}|T)}{2L}}\right] \geq 1 - \frac{2}{T}.$$

For $p \in \Delta^K$, let $\overline{p} \in \arg\min_{p' \in \mathcal{P}} \|p - p'\|_1$. Then, since $\mathcal{P}$ is a $\delta$-cover, we have $\|p - \overline{p}\|_1 \leq \delta$. Thus, due to Lemma 2, we have

$$|\text{NSW}(p, \widehat{\mu}) - \text{NSW}(p, \mu^*)| \leq \sum_{\mu \in \{\widehat{\mu}, \mu^*\}} |\text{NSW}(p, \mu) - \text{NSW}(\overline{p}, \mu)| + |\text{NSW}(\overline{p}, \widehat{\mu}) - \text{NSW}(\overline{p}, \mu^*)|$$

$$\leq 2N\delta + |\text{NSW}(\overline{p}, \widehat{\mu}) - \text{NSW}(\overline{p}, \mu^*)|.$$

Setting $\delta = \frac{1}{NT}$, we have

$$\Pr\left[\forall p \in \Delta^K : |\mathrm{NSW}(p, \widehat{\mu}) - \mathrm{NSW}(p, \mu^*)| \le \frac{2}{T} + \sqrt{\frac{N \log(|\mathcal{P}|T)}{2L}}\right] \ge 1 - \frac{2}{T}.$$

Next, we use the fact that

$$\mathrm{NSW}(p^*, \mu^*) - \mathrm{NSW}(\widehat{p}, \mu^*) \le \sum_{p \in \{p^*, \widehat{p}\}} |\mathrm{NSW}(p, \widehat{\mu}) - \mathrm{NSW}(p, \mu^*)|.$$

Hence,

$$\Pr\left[|\mathrm{NSW}(p^*, \mu^*) - \mathrm{NSW}(\widehat{p}, \mu^*)| \le \frac{4}{T} + \sqrt{\frac{2N \log(|\mathcal{P}|T)}{L}}\right] \ge 1 - \frac{2}{T}.$$

Next, we substitute $|\mathcal{P}| \le (1 + 2/\delta)^K \le (3/\delta)^K$, $\delta = \frac{1}{NT}$, and $L = \Theta\left(N^{\frac{1}{3}} K^{-\frac{1}{3}} T^{\frac{2}{3}} \log^{\frac{2}{3}}(NKT)\right)$, and then substitute the derived bound in Equation (2) to get the second regret bound. $\qquad\square$

## B.5 Proof of Lemma 4

*Proof.* Fix $t$. For $i \in [N]$, $j \in [K]$, and $\ell \in [t]$, let $\overline{v}_{i,j}^\ell$ denote the average reward to agent $i$ from the first $\ell$ pulls of arm $j$, and define $\overline{r}_j^\ell = \sqrt{\frac{2 \log(NKt)}{\ell}}$. Then, by Hoeffding's inequality, we have

$$\forall i \in [N], j \in [K], \ell \in [t] : \Pr\left[\left|\overline{v}_{i,j}^\ell - \mu_{i,j}\right| > \overline{r}_j^\ell\right] \le \frac{2}{(NKt)^4}.$$

By the union bound, we get

$$\Pr\left[\forall i \in [N], j \in [K], \ell \in [t] : \left|\overline{v}_{i,j}^\ell - \mu_{i,j}\right| \le \overline{r}_j^\ell\right] \ge 1 - \frac{2}{(NKt)^3}.$$

Because $n_j^t \in [t]$ for each $j \in [K]$, the above event implies our desired event $\mathcal{E}^t$. Hence, we have that $\Pr[\mathcal{E}^t] \ge 1 - 2/(NKt)^3 \ge 1 - 2/t^3$. $\qquad\square$

## B.6 Proof of Lemma 5

*Proof.* Conditioned on $\mathcal{E}^t$, we have $\left|\widehat{\mu}_{i,j}^t - \mu_{i,j}^*\right| \le r_j^t$ for each $j \in [K]$. In that case, it is easy to see that the upper bound from Lemma 3 becomes $N \cdot \sum_{j \in [K]} p_j \cdot r_j^t$. $\qquad\square$

## B.7 Proof of Lemma 6

*Proof.* Fix $p \in \Delta^K$. Fix $\delta > 0$, and let $\mathcal{P}$ be a $\delta$-cover of the policy simplex $\Delta^K$ with $|\mathcal{P}| \le (1 + 2/\delta)^K$ [44, p. 126].

Conditioned on $\mathcal{E}^t$ (i.e. $\left|\widehat{\mu}_{i,j}^t - \mu_{i,j}^*\right| \le r_j^t = \sqrt{\frac{2 \log(NKt)}{n_j^t}}, \forall i \in [N], j \in [K]$), we wish to derive a high probability bound on $|\mathrm{NSW}(p, \widehat{\mu}^t) - \mathrm{NSW}(p, \mu^*)|$. We can bound the deviation of $\mathrm{NSW}(p, \widehat{\mu}^t)$ from its expected value. However, unlike in the case of Explore-First, we cannot directly claim that the expected value is $\mathrm{NSW}(p, \mu^*)$ because, as we mentioned above, $\widehat{\mu}^t$ consists of random variables that may be correlated through the random variable $n^t = (n_1^t, \ldots, n_K^t)$ taking values in $[t]^K$. Thus, we need a more careful argument.

For $i \in [N]$, $j \in [K]$, and $\ell_j \in [t]$, let $\overline{v}_{i,j}^{\ell_j}$ denote the average reward to agent $i$ from the first $\ell_j$ pulls of arm $j$, and define $\overline{r}_j^{\ell_j} = \sqrt{\frac{2 \log(NKt)}{\ell_j}}$. Let $\ell = (\ell_1, \ldots, \ell_K) \in [t]^K$ and $\overline{v}^\ell = (\overline{v}_{i,j}^{\ell_j})_{i \in [N], j \in [K]}$. Each $\overline{v}_{i,j}^{\ell_j}$ is independent and satisfies $\mathbb{E}[\overline{v}_{i,j}^{\ell_j}] = \mu_{i,j}^*$. Since expectation decomposes over sums and products of independent random variables, we have $\mathbb{E}[\mathrm{NSW}(p, \overline{v}^\ell)] = \mathrm{NSW}(p, \mu^*)$.

**Evaluating conditional expectation:** We next argue that further conditioning on the high probability event $\mathcal{E}^t$ does not change the expectation by much. Formally,

$$\left| \text{NSW}(p, \mu^*) - \mathbb{E}[\text{NSW}(p, \overline{v}^\ell) | \mathcal{E}^t] \right|$$
$$= \left| \mathbb{E}[\text{NSW}(p, \overline{v}^\ell)] - \mathbb{E}\left[\text{NSW}(p, \overline{v}^\ell) | \mathcal{E}^t\right] \right|$$
$$= \Pr[\neg\mathcal{E}^t] \cdot \left| \mathbb{E}\left[\text{NSW}(p, \overline{v}^\ell) | \neg\mathcal{E}^t\right] - \mathbb{E}\left[\text{NSW}(p, \overline{v}^\ell) | \mathcal{E}^t\right] \right|$$
$$\leq \Pr[\neg\mathcal{E}^t] \leq \frac{2}{t^3} \leq \frac{2}{t}, \tag{3}$$

where the penultimate transition holds because NSW is bounded in $[0, 1]$, and the final transition is due to Lemma 4.

**Applying McDiarmid's inequality:** We first decompose $\overline{v}^\ell$ into $N$ random variables: for each $i \in [N]$, let $\overline{v}_i^\ell = (\overline{v}_{i,j}^\ell)_{j \in [K]}$. To apply McDiarmid's inequality, we need to analyze the maximum amount $c_i$ by which changing $\overline{v}_i^\ell$ can change $\text{NSW}(p, \overline{v}^\ell)$. Fix $i \in [N]$, and fix all the variables except $\overline{v}_i^\ell$. Conditioned on $\mathcal{E}^t$, each $\overline{v}_{i,j}^\ell$ can change by at most $2\overline{r}_j^{\ell_j}$. Hence, using Lemma 3, we have that $c_i \leq 2 \sum_{j \in [K]} p_j \cdot \overline{r}_j^{\ell_j}$. Now, applying McDiarmid's inequality, we have $\forall \ell \in [t]^K$ :

$$\Pr\left[\left|\text{NSW}(p, \overline{v}^\ell) - \mathbb{E}\left[\text{NSW}(p, \overline{v}^\ell) | \mathcal{E}^t\right]\right| \geq \epsilon \mid \mathcal{E}^t\right] \leq 2e^{\frac{-2\epsilon^2}{\sum_{i \in [N]} c_i^2}} \leq 2e^{\frac{-2\epsilon^2}{4N \cdot \left(\sum_{j \in [K]} p_j \cdot \overline{r}_j^{\ell_j}\right)^2}}.$$

Using Equation (3), and setting $\epsilon = \sqrt{2N \log(|\mathcal{P}| t^{K+3})} \cdot \sum_{j \in [K]} p_j \cdot \overline{r}_j^{\ell_j}$, we have that $\forall \ell \in [t]^K$:

$$\Pr\left[\left|\text{NSW}(p, \overline{v}^\ell) - \text{NSW}(p, \mu^*)\right| \geq \sqrt{2N \log(|\mathcal{P}| t^{K+3})} \cdot \sum_{j \in [K]} p_j \cdot \overline{r}_j^{\ell_j} + \frac{2}{t} \mid \mathcal{E}^t\right] \leq \frac{2}{|\mathcal{P}| t^{K+3}}.$$

Next, by union bound, we get

$$\Pr\left[\forall \ell \in [t]^K : \left|\text{NSW}(p, \overline{v}^\ell) - \text{NSW}(p, \mu^*)\right| \geq \sqrt{2N \log(|\mathcal{P}| t^{K+3})} \cdot \sum_{j \in [K]} p_j \cdot \overline{r}_j^{\ell_j} + \frac{2}{t} \mid \mathcal{E}^t\right]$$
$$\leq \frac{2}{|\mathcal{P}| t^3}.$$

Because $n_j^t \in [t]$ for each $j \in [K]$, we have

$$\Pr\left[\left|\text{NSW}(p, \widehat{\mu}^t) - \text{NSW}(p, \mu^*)\right| \geq \sqrt{2N \log(|\mathcal{P}| t^{K+3})} \cdot \sum_{j \in [K]} p_j \cdot r_j^t + \frac{2}{t} \mid \mathcal{E}^t\right]$$
$$\leq \frac{2}{|\mathcal{P}| t^3}.$$

**Extending to all policies in $\mathcal{P}$:** Using the union bound, we have that

$$\Pr\left[\forall p \in \mathcal{P} : \left|\text{NSW}(p, \widehat{\mu}^t) - \text{NSW}(p, \mu^*)\right| \leq \sqrt{2N \log(|\mathcal{P}| t^{K+3})} \cdot \sum_{j \in [K]} p_j \cdot r_j^t + \frac{2}{t} \mid \mathcal{E}^t\right]$$
$$\geq 1 - \frac{2}{t^3}.$$

**Extending to all policies in $\Delta^K$:** For $p \in \Delta^K$, let $\overline{p} \in \arg\min_{p' \in \mathcal{P}} \|p - p'\|_1$. Then, since $\mathcal{P}$ is a $\delta$-cover, we have $\|p - \overline{p}\|_1 \leq \delta$. Thus, due to Lemma 2, we have

$$\left|\text{NSW}(p, \widehat{\mu}^t) - \text{NSW}(p, \mu^*)\right| \leq \sum_{\mu \in \{\widehat{\mu}^t, \mu^*\}} \left|\text{NSW}(p, \mu) - \text{NSW}(\overline{p}, \mu)\right|$$
$$+ \left|\text{NSW}(\overline{p}, \widehat{\mu}^t) - \text{NSW}(\overline{p}, \mu^*)\right|$$
$$\leq 2N\delta + \left|\text{NSW}(\overline{p}, \widehat{\mu}^t) - \text{NSW}(\overline{p}, \mu^*)\right|.$$

Setting $\delta = \frac{1}{Nt}$, we have

$$\Pr\left[\forall p \in \Delta^K : \left|\mathrm{NSW}(p, \widehat{\mu}^t) - \mathrm{NSW}(p, \mu^*)\right| \leq \sqrt{2N \log(|\mathcal{P}| t^{K+3})} \cdot \sum_{j \in [K]} p_j \cdot r_j^t + \frac{4}{t} \,\middle|\, \mathcal{E}^t\right]$$

$$\geq 1 - \frac{2}{t^3}.$$

Substituting $|\mathcal{P}| \leq (1 + 2/\delta)^K \leq (3/\delta)^K$ with $\delta = \frac{1}{Nt}$ yields the desired bound. $\qquad\square$

## B.8 Proof of Theorem 2

*Proof.* Fix $t \in [T]$. Let $b^t$ denote the number of times Epsilon-Greedy performs exploration up to round $t$. Note that $\mathbb{E}[b^t] = \sum_{s=1}^{t} \epsilon^s \geq t\epsilon^t$, where the last step follows from the fact that $\epsilon^t$ is monotonically decreasing in both cases of the theorem. Let $\theta > 0$ be a constant such that $\epsilon^t \geq \theta \cdot t^{-1/3}$ in both cases of the theorem.

Define the event $\mathcal{B}^t \triangleq b^t \geq \gamma \cdot t\epsilon^t$, where $\gamma = 1 - 1/\theta$. Then, by Hoeffding's inequality, we have

$$\Pr[\neg\mathcal{B}^t] \leq e^{-2(1-\gamma)^2\theta^2 t^{1/3}} = e^{-2t^{1/3}} \leq e^{-\log t} = \frac{1}{t}. \tag{4}$$

Because the algorithm performs round-robin during exploration, conditioned on $\mathcal{B}^t$, we have that $n_j^t \geq \frac{b^t}{K} \geq \frac{\gamma \cdot t\epsilon^t}{K}$ for each arm $j$,[6] which implies $r_j^t \leq \sqrt{\frac{2K \log(NKt)}{\gamma \cdot t\epsilon^t}}$ for each $j$. Thus, conditioned on $\mathcal{B}^t$, we have

$$\forall p \in \Delta^K : \sum_{j \in [K]} p_j \cdot r_j^t \leq \max_{j \in [K]} r_j^t \leq \sqrt{\frac{2K \log(NKt)}{\gamma \cdot t\epsilon^t}}. \tag{5}$$

We are now ready to use the bounds from Lemmas 5 and 6. We focus on the event

$$\mathcal{C}_\alpha^t \triangleq \forall p \in \Delta^K : \left|\mathrm{NSW}(p, \mu^*) - \mathrm{NSW}(p, \widehat{\mu}^t)\right| \leq \alpha^t \cdot \sum_{j \in [K]} p_j \cdot r_j^t + \frac{4}{t}.$$

Conditioned on $\mathcal{E}^t \wedge \mathcal{H}^t$, note that $\mathcal{C}_\alpha^t$ holds for $\alpha^t = N$ due to Lemma 5, and for $\alpha^t = \sqrt{12NK \log NKt}$ due to Lemma 6.

Let $\widehat{p}^t \in \arg\max_{p \in \Delta^K} \mathrm{NSW}(p, \widehat{\mu}^t)$. We wish to bound the regret $\mathrm{NSW}(p^*, \mu^*) - \mathrm{NSW}(\widehat{p}^t, \mu^*)$ that Epsilon-Greedy incurs when performing exploitation in round $t$ by choosing policy $\widehat{p}^t$. Conditioned on $\mathcal{E}^t \wedge \mathcal{H}^t \wedge \mathcal{B}^t$, we have

$$\begin{aligned}
&\mathrm{NSW}(p^*, \mu^*) - \mathrm{NSW}(\widehat{p}^t, \mu^*) \\
&= \left(\mathrm{NSW}(p^*, \mu^*) - \mathrm{NSW}(p^*, \widehat{\mu}^t)\right) + \left(\mathrm{NSW}(p^*, \widehat{\mu}^t) - \mathrm{NSW}(\widehat{p}^t, \widehat{\mu}^t)\right) \\
&\quad + \left(\mathrm{NSW}(\widehat{p}^t, \widehat{\mu}^t) - \mathrm{NSW}(\widehat{p}^t, \mu^*)\right) \\
&\leq \sum_{p \in \{p^*, \widehat{p}^t\}} \left|\mathrm{NSW}(p, \mu^*) - \mathrm{NSW}(p, \widehat{\mu}^t)\right| \leq 2\alpha^t \sqrt{\frac{2K \log(NKt)}{\gamma \cdot t\epsilon^t}} + \frac{8}{t}, \tag{6}
\end{aligned}$$

where the penultimate transition holds because $\widehat{p}^t$ is the optimal policy under $\widehat{\mu}^t$, so $\mathrm{NSW}(p^*, \widehat{\mu}^t) \leq \mathrm{NSW}(\widehat{p}^t, \widehat{\mu}^t)$, and the final transition follows from Equation (5) and the fact that $\mathcal{E}^t \wedge \mathcal{H}^t$ imply $\mathcal{C}_\alpha^t$.

---

[6]Technically, $n_j^t \geq \lfloor \frac{b^t}{K} \rfloor$ for each arm $j$, but we omit the floor for the ease of presentation.

We are now ready to analyze the expected regret of Epsilon-Greedy at round $T$. We have

$$\mathbb{E}[R^T] = \sum_{t=1}^T \mathbb{E}[r^t] \le \sum_{t=1}^T \mathbb{E}\left[\epsilon^t \cdot 1 + (1 - \epsilon^t) \cdot \left(\text{NSW}(p^*, \mu^*) - \text{NSW}(\widehat{p}^t, \mu^*)\right)\right]$$

$$\le \sum_{t=1}^T \Bigg(\epsilon^t + \Pr\left[\mathcal{E}^t \wedge \mathcal{H}^t \wedge \mathcal{B}^t\right] \cdot \mathbb{E}\left[\text{NSW}(p^*, \mu^*) - \text{NSW}(\widehat{p}^t, \mu^*) \,\Big|\, \mathcal{E}^t \wedge \mathcal{H}^t \wedge \mathcal{C}_\alpha^t\right]$$

$$+ \Pr\left[\neg\mathcal{E}^t \vee \neg\mathcal{H}^t \vee \neg\mathcal{B}^t\right] \cdot 1\Bigg)$$

$$\le \sum_{t=1}^T \left(\epsilon^t + 2\alpha^t \sqrt{\frac{2K \log(NKt)}{\gamma \cdot t\epsilon^t}} + \frac{8}{t} + \frac{4}{t^3} + \frac{1}{t}\right),$$

where the final transition holds due to Equation (6), Lemma 4, Lemma 6, and Equation (4). Notice that we are using the fact that

$$\Pr[\mathcal{E}^t \wedge \mathcal{H}^t] = \Pr[\mathcal{E}^t] \cdot \Pr[\mathcal{H}^t|\mathcal{E}^t] \ge (1 - 2/t^3) \cdot (1 - 2/t^3) \ge 1 - 4/t^3.$$

To obtain the first regret bound, we set $\epsilon^t = \Theta\left(N^{\frac{2}{3}} K^{\frac{1}{3}} t^{-\frac{1}{3}} \log^{\frac{1}{3}}(NKt)\right)$ and $\alpha^t = N$, and obtain

$$\mathbb{E}[R^T] = \mathcal{O}\left(N^{\frac{2}{3}} K^{\frac{1}{3}} \log^{\frac{1}{3}}(NKT) \sum_{t=1}^T t^{-\frac{1}{3}}\right) = \mathcal{O}\left(N^{\frac{2}{3}} K^{\frac{1}{3}} T^{\frac{2}{3}} \log^{\frac{1}{3}}(NKT)\right).$$

For the second regret bound, we set $\epsilon^t = \Theta\left(N^{\frac{1}{3}} K^{\frac{2}{3}} t^{-\frac{1}{3}} \log^{\frac{2}{3}}(NKt)\right)$ and $\alpha^t = \sqrt{12NK \log(NKt)}$, and obtain

$$\mathbb{E}[R^T] = \mathcal{O}\left(N^{\frac{1}{3}} K^{\frac{2}{3}} \log^{\frac{2}{3}}(NKT) \sum_{t=1}^T t^{-1/3}\right) = \mathcal{O}\left(N^{\frac{1}{3}} K^{\frac{2}{3}} T^{\frac{2}{3}} \log^{\frac{2}{3}}(NKT)\right).$$

Note that in both cases, we omit the $\mathcal{O}(1/t)$ and $\mathcal{O}(1/t^3)$ terms because they are dominated by the $\mathcal{O}(1/t^{1/3})$ term. In both cases, we use Proposition 1 at the end. $\qquad\square$