# OpenReview forum: "Fair Algorithms for Multi-Agent Multi-Armed Bandits"
_NeurIPS.cc/2021/Conference — NeurIPS 2021 Poster_

### Official Review · Reviewer_mA86 · 2021-06-30

**Rating:** 6
**Confidence:** 4

**Summary:**

In this paper, the authors propose to minimize the Nash Social Welfare (NSW) function -- the product of utilities, in a multi agent multi-armed bandit problem. The NSW strikes a balance between utilitarian and egalitarian optimization in a multi agent framework. The product of utility maximization is a log-concave optimization for which they propose three algorithms -- Explore-then-commit, Epsilon-Greedy, and UCB.  They establish regret upper bounds for the proposed algorithms, with UCB having the best \tilde{O}(\sqrt{T}) regret.

**Limitations And Societal Impact:**

This work is theoretical in nature. It deals with maximizing the Nash Welfare objective that is established to promote fairness in the literature. Thus, I think this paper will not have negative societal impact.

**Main Review:**

Pros

- The maximization of the NSW objective in a multi-agent setting promotes fairness consideration for the multi-agent bandits problem.

- Technically, the bandit optimization problem is a special log-concave for which this paper proposes the first \tilde{O}(\sqrt{T}) regret bound (to the best of my knowledge).  The authors also discuss why the existing results will not provide such a regret bound (see drawbacks for improving this).

- The writing is clear and easy to follow (except for a few aspects described shortly). The proof sketches are insightful.

Drawbacks and Suggestions

- The key drawback of the paper is the centralized optimization in the multi-agent setting. Maybe the authors can discuss how the agents are incentivized to submit truthful feedback to the centralized arbiter.

- Can you provide the details of the zooming dimension  (as per Kleinberg et al.) computation . Also, why the tree-search approach in  'Multi-scale exploration of convex functions and bandit convex optimization' not applicable here?

- The related works in the no-regret learning in non-cooperative games is not discussed properly. Please add these works. E.g. 'Bandit Learning in Concave N-Person Games' - Bravo et al. and the references therein.

- Provide a proof sketch of Lemma 6 in the main paper. Can the union bound on the possible number of pulls be avoided there? Also what is the slackness added by taking such a union bound?

- Discussions on when Lemma 5/Lemma 6 should be applied depending on N,K, and T will be useful. In particular, can we adaptively set \alpha_t in UCB by switching from N to O(\sqrt{NKlog(NKt)}) (or the opposite) to improve the regret bound?

**Time Spent Reviewing:**

5

---

> ### Author Response · Authors · 2021-08-11
> **Author Response**
>
> Thank you for your review and for pointing out the relevant literature. We will discuss this in the camera ready version.
>
> **Regarding truthfulness:** In our work, we consider settings where the rewards to the agents are publicly observable and focus on analyzing how we can combat the lack of full information to achieve maximum fairness. In other words, our work falls under social choice rather than mechanism design. That said, considering the impact of agents strategically revealing (parts of) their observed rewards would be an interesting direction for future work, akin to what is studied in https://arxiv.org/abs/1706.09060.
>
> **Regarding bandit convex optimization:** The main reason that we cannot apply this approach is because the Nash social welfare (NSW) is not a convex/concave function. While log(NSW) is concave and therefore can be approximately maximized using such techniques, regret guarantees in terms of log(NSW) do not translate to meaningful regret guarantees in terms of NSW. We have described this in Section 1.2.
>
> **Regarding Lemma 6:** In our current analysis, we do not see an immediate way to avoid the union bound you mentioned. Due to this union bound, our definition of $\epsilon$ in the proof of Lemma 6 uses the term $t^{K+3}$ instead of just $t^3$. This slackness translates to an additional multiplicative factor of $\sqrt{K}$ in our regret bound.
>
> **Regarding the choice of $\alpha^t$:** While it seems possible to adaptively change $\alpha^t$ and still derive a meaningful regret bound, note that the bounds achieved with the two values of $\alpha^t$ offer a tradeoff between the exponents of $N$ and $K$. Hence, the “right” choice of $\alpha^t$ can be made up front depending on which of $N$ and $K$ is smaller: use $\alpha^t = N$ if $N \ll K$ and $\alpha^t = \sqrt{12NK\log(NKt)}$ otherwise. There does not seem to be any benefit in making this choice adaptively.

---

> > ### Comment · Reviewer_mA86 · 2021-08-11
> > **Response**
> >
> > Thanks for the detailed response. I feel most response are reasonable except for: the omission of related works in the no-regret learning in non-cooperative games.
> >
> > For Tree search based approach (point 2 in drawback), I apologize for providing the wrong reference. I wanted to refer to the following where *convexity is not assumed*.
> >
> > S. Bubeck, R. Munos, G. Stoltz, and Szepesvári C. X-armed bandits. Machine Learning Research,
> > 12:1587–1627, 2011.
> >
> > Why can we not apply the methods therein? Keywords for finding related works are 'Monte-Carlo Tree Search' and 'Blackbox optimization'.  There also is Gaussian Process based approaches for general function. E.g.
> >
> > E. Brochu, V. M. Cora, and N. de Freitas. A tutorial on bayesian optimization of expensive
> > cost functions, with application to active user modeling and hierarchical reinforcement learning.
> > Technical report, University of British Columbia, 2010

---

> > > ### Author Response · Authors · 2021-08-14
> > > **Response to Response**
> > >
> > > Thank you for your response. We apologize for missing the part of your review asking about the calculation of the zooming dimension, which is related to the near-optimality dimension from the new reference you suggested.
> > >
> > > To see why the zooming dimension for our problem is $\Omega(k)$, consider an instance with two agents where $\mu^*_{1,1}=1, \mu^*_{1,j} = 0$ for all $j > 1$, $\mu^*_{2,1} = 0$, and $\mu^*_{2,j} = 1$ for all $j > 1$. The optimal NSW is $1/4$ and the set of policies $p$ with NSW between $\delta$ and $2\delta$ less than the optimal is characterized by $|p_1-1/2| \in [\sqrt{\delta},\sqrt{2\delta}]$ and $\sum_{j > 1} p_j = 1-p_1$. This region has volume $\Omega(\sqrt{\delta})$, whereas a ball of diameter $\delta/8$ in our $K$-simplex of policies has volume roughly $O(\delta)^{K-1}$, which leads to a zooming dimension of $\Omega(K)$. As the regret bound of Kleinberg et al. with zooming dimension $d$ is $\tilde{O}(T^{(d+1)/(d+2)})$, it is much worse than our $\tilde{O}(\sqrt{T})$ bound. The key challenge here is that NSW can sometimes admit a huge set of near-optimal policies.
> > >
> > > A similar argument also holds for the related quantity, near-optimality dimension, from your suggested reference, where, essentially, one can use a more general dissimilarity measure $\ell$ (indeed, in case of a metric, the near-optimality and zooming dimensions are equal). The key challenge of having to pack/cover a region of volume $\Omega(\sqrt{\delta})$ with balls of volume $O(\delta)^{K-1}$ remains.
> > >
> > > As promised, we will significantly expand our related work section to discuss these approaches in detail and what they imply for our problem, including the other lines of work you pointed out, e.g., Gaussian processes and no-regret learning in non-cooperative games.

---

> > > > ### Comment · Reviewer_mA86 · 2021-08-18
> > > > **Response**
> > > >
> > > > Thank you for the clarification. This improves the impression I have about the paper. However, I will stick to my score as NSW, although important in fairness, is a very specific function of interest in the bandit optimization literature.

---

### Official Review · Reviewer_qaWv · 2021-07-14

**Rating:** 6
**Confidence:** 4

**Summary:**

This paper studies a multi-armed bandits problem with multiple agents. Arms have agent specific reward distributions associated with them. An arm is pulled at each step and each agent receives a reward sampled from their corresponding agent specific reward distribution for this arm. The goal of the learner is to find a distribution over arms that maximizes an appropriate aggregate notion of rewards earned by all agents – the Nash welfare.

The paper develops variants of three standard algorithms – one that explores for a fixed number of rounds and then exploits (Explore-First), an epsilon-greedy style algorithm, and a UCB style algorithm, to minimize a notion of regret which measures the difference between the Nash welfare of the solution at time t and the optimal Nash welfare. The paper derives upper bounds on the expected regret of the algorithm in each case. For UCB, the upper bound matches the lower bound (in the appendix) in terms of its dependence on parameter T (time horizon).

**Limitations And Societal Impact:**

The paper discusses various limitations such as possible sub-optimal dependence on N and K, and difficulty in optimising the objective required by UCB.

**Main Review:**

Originality – This appears to be the first work that studies Nash welfare maximization in a multi-agent multi-armed bandits setting and obtains a \sqrt{T} regret bound. The paper does a good job of highlighting additional challenges posed by this setup. For instance, the mean estimates of the agents (for a given arm) are not independent as they all depend on the number of times that arm was pulled, which is same for all agents.

Quality –
1.	The theoretical results in the paper are fairly intuitive and appear to be sound. While the upper bound for UCB matches the lower bound in terms of T, the dependence on other parameters (K: #arms, N: #agents) appears to be suboptimal.
2.	Although the paper argues that “traditionally, the focus is on optimizing the exponent of T rather than that of K”, I do not completely agree with the statement, especially in the context of the current paper where N x K may be very large.
3.	A stronger lower bound in terms of K and N would have helped in understanding the quality of the current upper bounds. Even empirical results will provide additional information about the tightness of the current upper-bound.
4.	Combining the fact that the upper bound for UCB may be very loose (in terms of N and K) with the fact that it will be inefficient to implement, makes the assessment of the utility of the algorithm hard.

Clarity – The paper is very well written and easy to follow.

Significance – See point 4 under Quality. This makes it hard to argue for the significance of the current work. Having said that, I agree that the paper takes an important first step for an interesting problem.

Strengths – Very well written, the regret of UCB has optimal dependence in terms of T

Weaknesses – Unclear about the tightness of the bounds in terms of N and K (see details above)

**Update**: Dear Authors, thank you for your response. I agree that $T \geq K$ is a harmless assumption. However, the same is not true for $T \geq N$, and hence not true for $T \geq N \times K$ (am I missing something here?). I still believe that this is worth exploring, but the paper is undoubtedly a good first step in this direction. My score remains unchanged.

**Time Spent Reviewing:**

5

---

> ### Author Response · Authors · 2021-08-11
> **Author Response**
>
> Thank you for your review.
>
> First, note that $T \ge K$ is a harmless assumption. It is difficult to provide meaningful regret guarantees if one cannot even simple each arm once.
>
> In many real-world applications, decisions need to be made repeatedly and frequently; take, for instance, the recommender systems example we suggested to Reviewer dz9B. In such applications, it is reasonable to assume that $T$ is much larger than $N$ or $K$.
>
> However, you are correct that there are also data-scarce environments in which $T$ may be of the order of $N$, $K$, or $N \times K$. Hence, teasing out the optimal dependence on $N$ and $K$, in addition to $T$, is definitely an interesting direction for future work, as we mention in Section 6.

---

### Official Review · Reviewer_2Wto · 2021-07-15

**Rating:** 7
**Confidence:** 4

**Summary:**

This paper studies multi-agent multi-armed bandits under heterogeneity (where arm means vary across users), with the objective to optimize Nash social welfare to ensure fair allocation of pulls in the obtained policy. The authors present variants of explore-first, epsilon-greedy, and upper confidence bound strategies to achieve fair decision-making. Each algorithm obtains "fair" regret matching the standard counterpart, along with a $ \Omega(\sqrt{T})$ lower bound in the Appendix. The central technical (theoretical) contribution is a concentration result on the fluctuation of NSW that exploits the Lipschitz property of the objective coupled with a covering argument, which is then utilized subsequently to derive regret bounds for each of the proposed algorithms.

**Limitations And Societal Impact:**

The major limitations of the work as is are the "centralized" nature of the proposed policy and the intractability of the UCB algorithm presented. The authors should discuss this in more detail in the updated version.

Given that the paper discusses fairness in a technical sense in context of online learning, the authors have done a good job of addressing the societal impact of their work, and I do not foresee any additional negative societal impact of this work.

**Main Review:**

Originality and Quality: I like the topic and contributions of the paper. Fairness in multi-armed bandits is typically concerned with fairness across arms, and multi-agent decision-making typically involves purely cooperative objectives, which suffer from the _tyranny of the majority_ effect as the authors have rightly pointed out. The paper does present the feasibility of NSW as an objective for bandit algorithms and analyzes popular single-agent algorithms, and I feel that it serves as a good contribution to the development of more work in this area. However, while the paper does address multi-agent decision-making, the algorithms and setup are essentially "single-agent" in the sense that one controller decides the policy for all agents. From a theoretical and algorithmic perspective, this may appear to be a necessary first-step simplification, but most deployment scenarios for multi-agent decision-making do not allow for such synchronicity, which is something the authors should discuss.

The previous remark is related to the broader concern around neglecting quite a vast and fast-growing literature around multi-agent multi-armed bandits in more realistic, distributed environments [https://arxiv.org/abs/1606.00911, https://scholar.harvard.edu/files/shahin/files/icassp_2017.pdf, https://arxiv.org/abs/2008.06220, and references therein], where heterogeneity and delay in communication ensure that the algorithms proposed are "multi-agent" in the true sense of the term, rather than a single-agent controlling a more nuanced objective than cumulative reward. In this regard, this work is also related to multi-objective bandit optimization [https://people.eecs.berkeley.edu/~kandasamy/pubs/pariaUAI19mobo.pdf, https://arxiv.org/abs/1905.12879] where a measure of performance employed is Pareto optimality, instead of cumulative reward. Discussing these approaches in context with the proposed research will go a long way in shaping further research into practical algorithms in real-world settings, moving beyond regret bounds that are effectively "single-agent".

Going through the technical parts, I feel like the previous revisions have ironed out the remaining technical issues, although I am curious to know if Lemma 6 can perhaps be tightened more by considering a smaller class of policies than the entire K-simplex, following a similar line of argument for UCB policies as done in https://arxiv.org/abs/1907.05388, for example.

Clarity: The paper is well-written and easy to follow.

Significance: Fairness in online learning is an important topic of interest to the NeurIPS community.

In summary, after placing this work in context to the work in distributed multi-agent decision-making, I feel that it would be a good contribution to the conference.

**Time Spent Reviewing:**

6

---

> ### Author Response · Authors · 2021-08-11
> **Author Response**
>
> Thank you for your review and for pointing out these relevant lines of literature.
>
> Given space limitations, we opted to discuss only the works most closely related to ours on a *technical* level. However, you are correct that the works you point out are close to ours on a conceptual level, and it is important to contextualize our work with respect to them. In the camera ready version, we will significantly expand our related work section, discussing the papers you point out and others. If need be, we can relegate the discussion of Explore-First to the appendix or replace the full UCB proof with a proof sketch.
>
> We would also like to point out that, while each agent actively participating in the arm-pulling is indeed a fascinating direction for future research, we do not view our work as just a “simplifying first step”. As mentioned in our response to Reviewer dz9B, there are plenty of real-world scenarios where collective decision-making simultaneously affects multiple users, and our work provides algorithms to help learn how to make such collective decisions fairly.

---

### Official Review · Reviewer_dz9B · 2021-07-24

**Rating:** 5
**Confidence:** 3

**Summary:**

The paper studies stochastic multi-agent multi-armed bandits where the reward distribution could be heterogeneous, i.e., the mean reward for agents on an arm could be different.  The authors consider Nash social welfare as the fairness criteria and aim to learn a distribution on arm plays that result in optimal Nash social welfare. Thress algorithms based on Explore-Then-Commit, Epsilon Greedy, and UCB are given and their performance is analyzed.

**Limitations And Societal Impact:**

Yes

**Main Review:**

Using Nash social welfare as a fairness criterion in multi-armed-bandits is interesting. However, I felt that setup and analysis are weak. My comments are:

1) Why do all the agents have to play the same arm in each round. If they can also play different arms, cannot they learn faster?

2. Line 219-220:  If $\hat{\mu}_{i,j}^t$ is obtained from $n_j^t$ and the rewards have bounded support, why cannot be Hoeffding's inequality be applied? What violates the hypothesis of Hoeffding's result?

3. The bound in Lemma 5 holds with some probability or it is a deterministic bound?'

4. Lemma 4: In the definition of $\Epsilon^t$, are parenthesis missed?

**Time Spent Reviewing:**

6

---

> ### Author Response · Authors · 2021-08-11
> **Author Response**
>
> Thank you for your review. Please find our answers to your comments below.
>
> 1. In our model, the agents do not “play”; they are passive users being affected by the arm-pulling. The arms are pulled by a central decision-making body, which wants to maintain fairness between the affected users. Collective decision-making that affects several users is commonplace in the real world. For example, a (non-personalized) recommender system may need to generate recommendations that are fair with respect to the preferences of users from different demographics; think of the movies recommended on an in-flight entertainment system.
>
> 2. There are several reasons why we cannot apply Hoeffding’s inequality. First, $n_j^t$, the number of times arm $j$ is pulled, is a random variable; to apply Hoeffding’s inequality, this needs to be a fixed number. Second, because arms are chosen based on past rewards, even conditioned on a particular realization of $n_j^t$, the rewards being averaged are not necessarily independent. For more details check out Section 1.3.1 of this book: https://arxiv.org/pdf/1904.07272.pdf.
>
> 3. The bound in Lemma 5 holds deterministically. That said, note that the lemma’s statement starts with “Conditioned on $\mathcal{E}^t$, …”. Hence, the equation in the lemma’s statement is only claimed to hold when the good event $\mathcal{E}^t$ occurs, which happens with high probability (as shown in Lemma 4).
>
> 4. We do not believe any parentheses are missing. Everything to the right of $\triangleq$ is the definition of event $\mathcal{E}^t$. We are unsure what may be causing the confusion.

---

> > ### Comment · Reviewer_dz9B · 2021-08-28
> > **Unchaged score**
> >
> > I looked into the rebuttals.
> >
> > The authors partly cleared my understanding of the paper. But still, some points are unclear which makes it a bit uncomfortable to increase the score. It is a standard step in bandits that to apply Hoeffding's inequality one conditions on the number of pulls or applies union bound. But that does not preclude the application of Hoeffding's inequality in the bounds. Also, it is unclear why the rewards are independent in their setup.
> >
> > Overall, I found the technical novelty of the fairness method based on Nash Bargaining to be limited.
> >
> > I keep my score unchanged.

---

> > > ### Author Response · Authors · 2021-08-30
> > > **Regarding the inapplicability of Hoeffding's inequality**
> > >
> > > Thank you for reading our response. Just to further clarify why Hoeffding's inequality cannot be applied even after conditioning on the number of pulls, let us borrow the example from Section 1.3.1 of the book (https://arxiv.org/pdf/1904.07272.pdf) that we referenced in our previous response.
> > >
> > > Consider the usual (single agent) bandit setting. Suppose some arm's reward is $1$ with probability $0.5$ and reward $0$ with the remaining probability $0.5$. Suppose we want to apply Hoeffding's inequality conditioned on pulling the arm $100$ times. For this, we need that the rewards obtained in these $100$ pulls of the arm be independent conditioned on the arm being pulled $100$ times. However, consider a bandit algorithm that only pulls the arm $100$-th time if the reward from the past $99$ pulls were all $0$. In this case, conditioned on pulling the arm $100$ times, the rewards from the first $99$ pulls are extremely correlated. Thus, Hoeffding's inequality cannot be applied.
> > >
> > > Note that if we knew that our algorithm pulls arms without looking at the past rewards, then we could indeed apply Hoeffding's inequality. This is precisely what we do to bound the rewards during the exploration phase of the Explore-First algorithm. However, we cannot do so for the other two algorithms.
> > >
> > > Hope this clarifies the issue further.

---

### Decision · Program_Chairs · 2021-09-27

**Decision:**

Accept (Poster)

**Comment:**

The paper considers a variant of the MAB problem where there are N agents who may perceive the "reward" from the K arms differently. The goal considered in this paper is to obtain a fair distribution over the arms in terms of Nash social welfare. This application is new, even if the algorithms and analysis do not seem to be that different from the standard MAB literature. The classical lower bound is also shown to hold in this case.